# Serial section microscopy image inpainting guided by axial optical flow

## ABSTRACT

Volume electron microscopy (vEM) is becoming a prominent technique in three-dimensional (3D) cellular visualization. vEM collects a series of two-dimensional (2D) images and reconstructs ultrastructures at the nanometer scale by rational axial interpolation between neighboring sections. However, section damage inevitably occurs in the sample preparation and imaging process, suffering from manual operational errors or occasional mechanical failures. The damaged regions present blurry and contaminated structure information, even local blank holes. Despite significant progress in single-image inpainting, it is still a great challenge to recover missing biological structures, that satisfy 3D structural continuity among sections. In this paper, we propose an optical flow-based serial section inpainting architecture to effectively combine the 3D structure information from neighboring sections and 2D image features from surrounding regions. We design a two-stage reference generation strategy to predict a rational and detailed intermediate state image from coarse to fine. Then, a GAN-based inpainting network is adopted to integrate all reference information and guide the restoration of missing structures, while ensuring consistent distribution of pixel values across the 2D image. Extensive experimental results well demonstrate the superiority of our method over existing inpainting tools.

## CCS CONCEPTS

• **Computing methodologies** → *Reconstruction*.

## KEYWORDS

Serial sectioning images; Image inpainting; Optical flow; Generative adversarial networks

## 1 INTRODUCTION

Volume electron microscopy (vEM) plays a critical role in understanding biological structures across scales, revealing biological complexity from the arrangement of organelles within cells, through the tissues composed of cellular communities, to the structural components of organisms [5]. vEM builds connections between two-dimensional (2D) imaging series and three-dimensional (3D) ultra-structures at the nanometer scale, according to high-precision 2D section alignment among an image stack and subsequent rational axial interpolation between neighboring sections. Analyzing

**Unpublished working draft. Not for distribution.**

the reconstructed volume, vEM enables researchers to comprehensively interpret the intricate topology of organelles within the crowded environments of a cell. However, section damage often occurs during the preparation and imaging process inevitably, preventing researchers from obtaining consistently high-resolution volumes [22]. It is largely due to manual operational errors and mechanical precision limitations, resulting in information missing in local regions and further reducing the resolution of reconstructed biological structures.

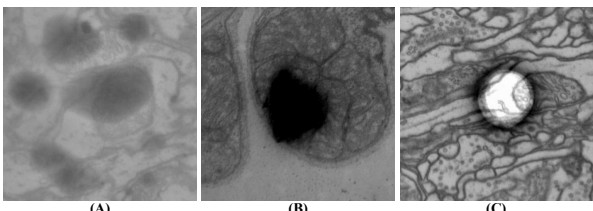

(A)      (B)      (C)

**Figure 1: Some common examples of damaged sections caused by (A) uneven staining, (B) imperfect serial sectioning, and (C) suboptimal microscope settings.**

The existence of section damage runs through the entire data acquisition process, involving heavy-metal staining, serial sectioning, and imaging. Different processes suffer specific image degradation patterns, but they all lead to invalid structural information (shown in Figure 1). Uneven staining caused by imperfect sample preparation greatly reduces the overall signal-to-noise ratio, and accompanying cloud-like sediments cover most of the biological structural areas. The blurred image greatly affects the contrast and pixel intensities of the reconstructed volume. Low-precision sectioning technique possibly generates sections with varying thicknesses, and even "erases" some regions of abundant structural information. The loss of section content undermines the foundation of high-resolution detailed structure reconstruction. Moreover, the erroneous or unstable microscope settings, such as probe dwelling time, electron beam, and so on, similarly lead to discontinuous or missing structures in a circular region. Under the collective influence of inevitable section damages, the reconstructed 3D volume may experience an inconsistent resolution or visualization scale across different across-sectional regions, severely interfering with the subsequent analysis and modeling. Although the above damages introduce a relatively large range of unreliable structural signals, abundant biological information remains in the section images. To achieve higher resolution and present clearer structures, missing information restoration in the damaged areas is becoming an urgent demand in vEM reconstruction.

Benefiting from coherent 3D attributes in serial sections, many works explore the utilization of information from neighbor sections to restore the damaged areas. CCPGAN [31] first proposes to "copy" the most relevant patches from both the damaged image and its neighboring image in the feature domain to recover

the missing regions. It relies heavily on the search for structurally similar regions, limiting its application scope to the repetition structure restoration with high-quality neighbor sections. Huang *et al.* [11] utilizes the interpolation result between two neighboring sections to guide the restoration of missing contents. The introduction of interpolation between sections preserves the axial continuity of biological structures and extends the application range. However, the blurred interpolation patches neglect the 2D structural consistency between the recovered content and undamaged parts. ACCP-GAN [23] designs a missing structure detection module to automatically determine the inpainted regions with arbitrary shapes. Transformer-based network enables the ability to capture structural similarity across input image stacks. Deng *et al.* [6] introduces a unified image inpainting framework to restore the artifacts caused by different image degradation. The interpolation result simply estimated between neighboring sections cannot provide clear and acceptable guidance for the subsequent inpainting module. In serial section inpainting, the 3D structural continuity and 2D image structural consistency are both the keys to rational and seamless restoration. Unfortunately, the axial trends of biological structures are non-trivial, without realistic ground truth images. The reference of interpolation results between neighboring sections is inevitably inconsistent with the damaged section, especially around the edge of missing areas. This raises a great challenge to bridge the gap between the reference image derived from 3D structural continuity and the damaged section of realistic 2D biological structures.

Here, we propose a GAN-based vEM section inpainting network called FlowInpaint, which estimates an intermediate state between neighboring sections to guide the restoration of damaged regions. Firstly, two neighboring images of the damaged section are fed into an initial reference module to derive a plausible intermediate state that satisfies the axial continuity of biological structures. Compared to raw neighboring images, this intermediate state exhibits more similar structures to those of the damaged section, thus significantly reducing the complexity of subsequent optical flow estimation. Then, the damaged section and the intermediate state are jointly input into the refined reference module, aiming to generate credible reference structures for the damaged regions. Finally, the GAN-based guided inpainting module employs a Unet-like architecture to extract multi-scale 2D image features from the reference image and raw neighbor sections. The abundant 2D structural features guide the rational restoration of missing structures, adhering to the 2D structural consistency around the missing areas. Moreover, a discriminator is used to ensure uniform pixel and noise value distribution, providing seamless and clear structure recovery in the inpainting regions. In summary, our main contributions are:

- We propose a novel vEM image inpainting framework called FlowInpaint to recover damaged regions in serial sections, considering both 3D structural continuity and 2D image structural consistency.
- We design a two-stage reference image generation strategy to estimate plausible biological structures in the inpainted regions from coarse to fine. The initial reference is interpolated between neighboring sections to preserve the axial 3D continuity. The subsequent refined reference module warps it to fit the realistic structures in the undamaged areas.

- We adopt a GAN-based image inpainting sub-network to generate final seamless results. This sub-network makes full use of multi-scale 2D image features from the reference and ensures the structural consistency around the damaged regions.
- We have conducted comprehensive experiments based on the CREMI datasets to demonstrate the superiority of Flow-Inpaint in serial section inpainting over existing methods. Extensive ablation and analysis results further indicate the stability of FlowInpaint in different imaging environments.

## 2 RELATED WORK

Video frame interpolation and image inpainting are the key techniques to exploring 3D structural continuity and 2D image structural consistency respectively.

### 2.1 Video Frame Interpolation

Video frame interpolation (VFI) aims to synthesize one or several frames in the middle of two adjacent frames of the original video. Traditional methods construct a series of explicit functions, such as polynomial functions [14], B-spline functions [2], and so on, to fit the changes of pixel values under the same plane coordinates. However, the biological structures perform elastic transformation in the axial direction, making the correspondences of plane coordinates unable to directly reflect the correspondences of structures.

With the development of deep learning, several convolutional neural networks (CNN) based VFI methods are presented to capture structural changes in-between the original frames. Long *et al.* [16] first attempt to use encoder-decoder architecture for the direct intermediate frame generation. Niklaus *et al.* [17] propose spatially-adaptive interpolation kernels to expand the receptive field without the excessive introduction of parameters. Cheng *et al.* [4] propose deformable separable convolution (DSepConv) to learn deformable offsets and masks, which can further extract features beyond the receptive field. These methods effectively restore the overall contour structures but exhibit a limited ability to recover more detailed or subtle structures. To trace the plausible motion of detailed structures, optical flow-based approaches estimate displacement for each pixel under the anti-folding constraints. Liu *et al.* [15] predict the 3D voxel flow to warp the input frames based on a trilinear sampling. DAIN [1] estimates the intermediate flow as a weighted combination of bidirectional flow and then warps the input frames using the adaptive warping layer. The above methods successfully estimate accurate and acceptable motion states of two input frames even under large displacement. However, in the serial section image inpainting, the missing information areas greatly influence the estimation of optical flow, making it difficult to estimate a reasonable deformation field directly from neighboring sections to the damaged area.

### 2.2 Image Inpainting

Existing image inpainting approaches can be classified into sequential-based algorithms and deep-learning based algorithms. Sequential-based algorithms [7] search the best-matching patches of damaged areas from the surrounding areas or given image database and "paste" them into the appropriate positions. By performing patch-wise computations, sequential-based approaches effectively

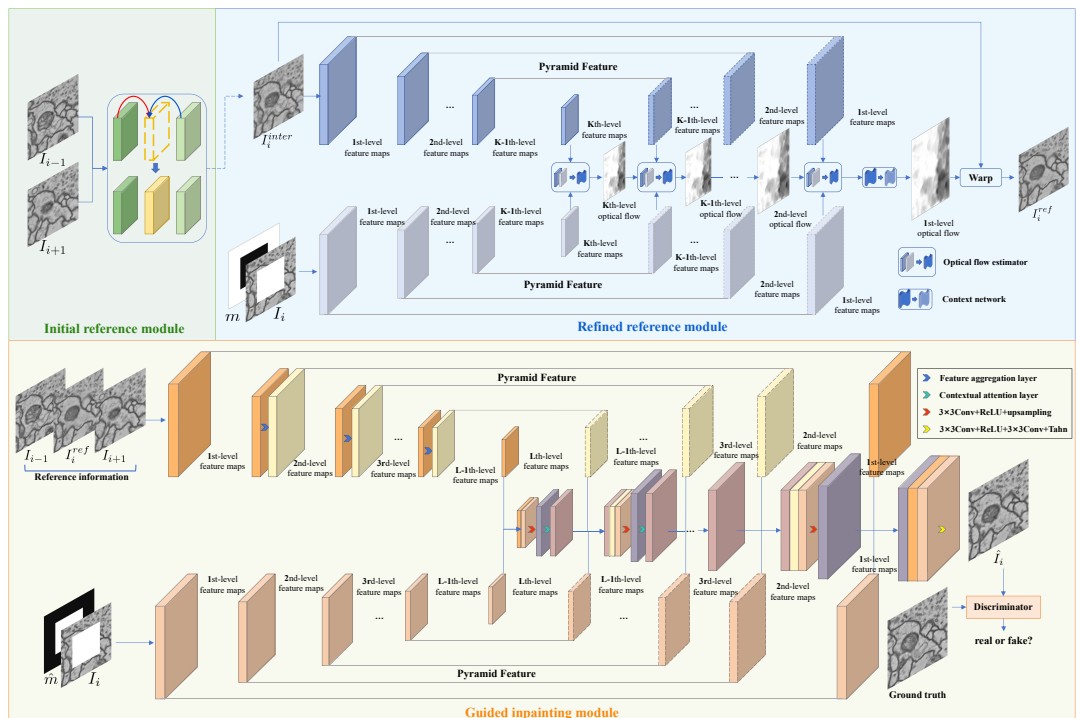

**Figure 2: The pipeline of FlowInpaint. Our model consists of the following three modules: (A) an initial reference module, (B) a refined reference module, and (C) a guided inpainting module. The initial reference module estimates a intermediate state image $I_i^{inter}$ between two neighboring images $I_{i+1}$ and $I_{i-1}$. Then the refined reference module utilizes a multi-scale architecture to gradually predict the optical flow $\overrightarrow{f}(inter_i, i)$ from the reference $I_i^{inter}$ to damaged image $I_i$. Finally, the guided inpainting module aggregates the information from three reference images to fill the missing areas and generate the final inpainted result $\hat{I}_i$. In the training stage, a hybrid discriminator is adopted to ensure the consistency between inpainted content and known regions.**

transform the complex image inpainting task into a simple database retrieval problem, whose key is the construction of similarity measurement between patches.

Different from sequential-based algorithms, deep learning-based methods have a strong ability to extract complex texture features and establish pixel-level correspondences in the feature domain [26]. Recently, the generative adversarial network (GAN) [9] has emerged as a promising paradigm for clear and rational image content recovery. The introduction of discriminators effectively ensures the consistent distribution of signal and noise values across the final restored image. Inspired by the attention mechanism, contextual attention [27] is proposed to depict the patch-based correlation between missing regions and background areas explicitly. Researchers explore designing different feature extraction architectures to provide comprehensive feature descriptions of image patches or pixels. AOT-GAN [29] adjusts the dilation rates of different dilated convolution layers to expand or shrink the receptive fields, offering both large-range and small-range structural features. Quan *et al.* [19] combine respective advantages of the convolution with small receptive fields and large-scale attention mechanism to globally and locally restore the missing information. Zhang *et al.* [32] introduce semantic priors from specific pretext tasks to image inpainting. Existing deep-learning based algorithms have exhibited powerful

abilities to preserve 2D structural consistency between missing and background areas in natural image inpainting. However, in serial section image inpainting, a reliable and high-resolution reconstruction of local structures depends crucially on the 3D axial structural continuity. Wang *et al.* [25] fuse useful features of neighboring images to reconstruct the intermediate image. Without information from neighboring sections, the image inpainting from 2D images alone cannot meet the requirements of downstream analysis tasks.

## 3 METHODOLOGY

In this section, we provide a detailed introduction to the FlowInpaint architecture as shown in Figure 2, which consists of three main modules, i.e., (a) the initial reference module, (b) the refined reference module, and (c) the guided inpainting module. The two neighboring images $I_{i-1}$ and $I_{i+1}$ of the damaged images $I_i$ are first input into the initial reference module to estimate an acceptable intermediate state result $I_i^{inter}$. The $I_i^{inter}$ is interpolated by considering bidirectional optical flows between $I_{i-1}$ and $I_{i+1}$ and exhibits more similar structures to the $I_i$. Then the $I_i^{inter}$, together with the $I_i$ and a binary mask $m$, is fed into the refined reference module to estimate the pixel displacement filed from $I_i^{inter}$ to $I_i$. The Unet-like network helps to extend the deformation from the

undamaged regions to the missing areas across scales. The final reference image $I_i^{ref}$ is acquired by warping the $I_i^{inter}$ with the estimated pixel displacement field. Finally, the GAN-base guided inpainting module accepts the $I_i^{ref}$, $I_{i-1}$ and $I_{i+1}$ as guidance information to restore the missing structures in $I_i$. The discriminator forces the 2D structural consistency around the missing regions while the reference image provides 3D axial continuity constraints of the internal structures.

## 3.1 Initial Reference Module

The initial reference module accepts two neighboring images $I_{i-1}$ and $I_{i+1}$ of damaged section $I_i$ to generate a stable intermediate state $I_i^{inter}$. Classical interpolation techniques use functions to fit the changes of pixel values of corresponding positions, unable to trace the transformation of structures. Here, we adopt an optical flow-based method raised by González-Ruiz $et\ al.$ [8] to depict the changes of axial structures between inputs $I_{i-1}$ and $I_{i+1}$. This module first estimates two optical flows $\overrightarrow{f}(i-1,i+1)$ from $I_{i-1}$ to $I_{i+1}$ and $\overrightarrow{f}(i+1,i-1)$ from $I_{i+1}$ to $I_{i-1}$. Considering the structural continuity, we assume that the biological structures are transformed smoothly along the axial direction. Therefore, we can acquire two candidate intermediate states by simply weighting on the estimated optical flows as follows:

$$I_{i,i-1}^{inter} = \mathbf{W}(I_{i-1}, \alpha \overrightarrow{f}(i-1,i+1)), \qquad (1)$$

$$I_{i,i+1}^{inter} = \mathbf{W}(I_{i+1}, (1-\alpha) \overrightarrow{f}(i+1,i-1)), \qquad (2)$$

which $\mathbf{W}(i,f)$ warps the image $i$ with a optical flow $f$, and $\alpha$ is a weighting coefficient representing the transformation degree. In this paper, we uniformly set the $\alpha$ to 0.5, assuming that all sections have the same thickness. Each candidate image exhibits the trend of structure transformation from the source image effectively without folding phenomenon. Finally, we fuse the two candidates to generate a bidirectional accessible state using the following weighting strategy:

$$I_i^{inter} = \alpha I_{i,i-1}^{inter} + (1-\alpha) I_{i,i+1}^{inter}. \qquad (3)$$

## 3.2 Refined Reference Module

The refined reference module uses the initial intermediate state image $I_i^{inter}$, the damaged image $I_i$ and a binary mask $m$ to estimate the optical flow $\overrightarrow{f}(inter_i, i)$ from $I_i^{inter}$ to $I_i$. The binary mask $m$ indicates the range of missing information, where value 1 represents valid content and value 0 marks the missing region. This module adapts from an optical flow estimation network PWC-Net [20], originally designed for optical flow estimation between two complete images according to multi-scale and step-wise decomposing pixel displacement fields.

This module involves two main components: a multi-scale feature extractor and cascaded optical flow estimators. The multi-scale feature extractor employs an Unet-like architecture to encode images $I_i^{inter}$ and $I_i$ along with $m$ into two corresponding pyramidal feature sets $\{I_i^{inter\,(k)}\}_{k=1}^K$ and $\{I_i^{(k)}\}_{k=1}^K$. With the increase of the layer, the feature maps keep being abstracted and filled with more

context information. This pyramidal architecture facilitates the capture of the changes in biological structures from rough trends to detailed warping. The cascaded optical flow estimators $\{\mathbf{E}_k(\cdot)\}_{k=1}^K$ are embedded into the hierarchical feature extractor, aiming to decompose and estimate large motions of biological structures at different scales. Accepting the $k$th feature maps $I_i^{(k)}$ and $I_i^{inter\,(k)}$, the $k$th optical flow estimator $\mathbf{E}_k(\cdot)$ first measures feature correlation based on the $\overrightarrow{f}_{(k+1)}(inter_i, i)$, outputted from the $(k+1)$th optical flow estimator. The feature correlation $C_{(k)}$ explicitly points out the regions or directions of the fitting target for the current level estimator, which is calculated as follows:

$$C_{(k)} = \frac{1}{N} I_i^{(k)T} W(I_i^{inter\,(k)}, \overrightarrow{f}_{(k+1)}(inter_i, i)), \qquad (4)$$

where $T$ is the transpose operator, $N$ is the length of the feature vector $I_i^{(k)}$, and $Up(\cdot)$ represents the ×2 upsampling. According to the identification of low feature correlation, the estimator forces the sub-network $\mathbf{E}_k(\cdot)$ to pay more attention to unregistered areas, as shown in Figure 3(A). The optical flow $\overrightarrow{f}_{(k)}(inter_i, i)$ at the $k$th level is estimated by integrating all extracted information:

$$\overrightarrow{f}_{(k)}(inter_i, i) = Up(\mathbf{E}_k(C_{(k)}, I_i^{(k)}, \overrightarrow{f}_{(k+1)}(inter_i, i), m_k)), \quad (5)$$

where the binary mask $m_k$ is downsampled from $m$ to match the current feature size. Finally, we utilize an additional context network $\mathbf{R}(\cdot)$ to refine the estimated flow $\overrightarrow{f}_{(1)}(inter_i, i)$ of the last level before upsampling, as shown in Figure 3(B).

This context network $\mathbf{R}(\cdot)$ uses a series of dilated convolution kernels with different dilated rates to enlarge the receptive fields, thus further smoothing the final pixel displacement field $I_i^{ref}$:

$$I_i^{ref} = \mathbf{W}(I_i^{inter}, Up(\mathbf{R}(\overrightarrow{f}_{(1)}(inter_i, i)))). \qquad (6)$$

## 3.3 Guided Inpainting Module

The guided inpainting module integrates all guidance information including the neighboring images $I_{i-1}$, $I_{i+1}$, and the refined reference images $I_i^{ref}$ to recover the missing regions of damaged image $I_i$. To ensure the consistent distribution of pixel values, we adopt a GAN-based architecture consisting of an attention-based generator and a patch-based discriminator, as shown in Figure 2.

The generator first constructs two multi-scale feature pyramids $\{\eta_1^l\}_{l=1}^L$ and $\{\eta_2^l\}_{l=1}^L$ from the guidance images $\{I_{i-1}, I_{i+1}, I_i^{ref}\}$ and $I_i$ using a series of convolution layers (shown in Figure 2). The accompanying binary mask $\hat{m}$, equaling $1 - m$, helps the network concentrate on the location of missing information. Notably, compared to the pyramidal extraction branch on the damaged image $I_i$, we add several extra feature aggregation layers (FAL) in the guidance image branch to extract and aggregate richer contextual features. Inspired by the atrous spatial pyramid pooling (ASPP) strategy [3], the FAL consists of four dilated convolutions $g_r(\cdot)$ ($r = 1, 2, 4, 8$) with different dilation rates, providing an extended characterization of local structures at different receptive fields. A large dilation rate enables the convolution layer to capture a wide range of pixel information, but also easily results in low resolution and content loss. To fully utilize the advantages of dilated convolutions, the FAL aggregates the results of four dilated convolutions

using a single convolution layer $g(\cdot)$:

$$\eta_1^l = g(g_1(\widetilde{\eta}_1^l) \oplus g_2(\widetilde{\eta}_1^l) \oplus g_4(\widetilde{\eta}_1^l) \oplus g_8(\widetilde{\eta}_1^l)), \tag{7}$$

where $\oplus$ is a concatenation operation, $\widetilde{\eta}_1^l$ represents the feature at the $l$th level. For the level $l$, the $l$th feature attention layer accepts the

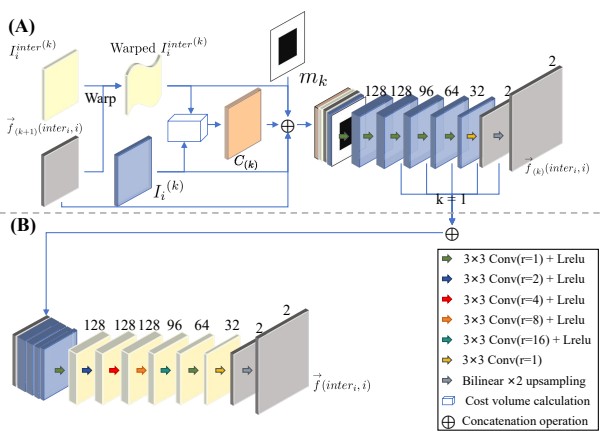

**Figure 3: An illustration of (A) the optical flow estimator $E_k(\cdot)$ on the top and (B) the context network $R(\cdot)$ on the bottom. The context network $R(\cdot)$ refines the output flow $\vec{f}_{(1)}(inter_i, i))$ of the optical flow estimator $E_1(\cdot)$ to offer the target flow $\vec{f}(inter_i, i))$.**

image features $\eta_1^l, \eta_2^l$ and the attention feature $\Psi^{l+1}$ from the former level $l+1$ to generate the fused feature. The feature attention layer consists of a feature fusion layer $\hbar(\cdot)$, integrating guidance features, and a contextual attention layer $att(\cdot)$, "copying" the similar patches in the feature domain. The contextual attention mechanism has been introduced to ensure better results in multi-scale [24]. The contextual attention layer $att(\cdot)$ [27] is used to model the patch $(3 \times 3)$ correlation between known regions $K^l$ and missing regions $R^l$ in the fused feature at the $l$th level. Let $k_i^l$ denotes the $i$th patch in the known region and $r_j^l$ denotes the $j$th patch in the missing region, the patch affinity $\tilde{s}_{i,j}^l$ is measured by the cosine similarity:

$$\tilde{s}_{i,j}^l = \langle \frac{k_i^l}{\|k_i^l\|_2}, \frac{r_j^l}{\|r_j^l\|_2} \rangle, \tag{8}$$

and the attention score $S_{i,j}^l \in S$ is obtained by the softmax operation:

$$S_{i,j}^l = \frac{exp(\tilde{s}_{i,j}^l)}{\sum_j exp(\tilde{s}_{i,j}^l)}. \tag{9}$$

Based on the feature similarity, the unknown regions $r_j^l$ in the fused feature at the $l$th level are updated by weighting the content of known patches:

$$\tilde{r}_j^l = \sum_{i=1} S_{i,j}^l k_i^l. \tag{10}$$

So the attention feature $\Psi^l$ is represented as:

$$\Psi^l = att(Up(\hbar(\eta_1^l \oplus \eta_2^l \oplus \Psi^{l+1})), \hat{m}). \tag{11}$$

Finally, the attention feature $\Psi^2$ is processed through a convolution block $d(\cdot)$ for the generation of the inpainting result $\hat{I}_i$ as follows:

$$\hat{I}_i = I_i \odot (1-m) + d(\eta_1^1 \oplus \eta_2^1 \oplus \Psi^2) \odot m, \tag{12}$$

where $\odot$ is pixel-wise multiplication.

The convolution architecture prefers to generate blurred visual results. To improve the perceptual quality of the inpainted image $\hat{I}_i$, we adopt PatchGAN [12] as the discriminator, to ensure the consistent distribution of pixel values and the continuity of structures around the missing areas. The discriminator consists of a global context discriminator network and a local context discriminator network, judging the rationality of restored information at different scales. The outputs of the two discriminators are concatenated together to finally predict a value within the range of $[0, 1]$, representing the probability that the input image is real or fake.

### 3.4 Loss Function

FlowInpaint is trained in a two-stage strategy with the same ground-truth image $x$, so as the loss functions.

In the first stage, we train the refined reference network, and the loss function $\mathcal{L}_C$ is defined by collecting pyramidal losses $\mathcal{L}_C^{(k)}$ across all levels as follows:

$$\mathcal{L}_C = \sum_{k=1}^{K} w_k \mathcal{L}_C^{(k)}. \tag{13}$$

At each level $k$, the loss $\mathcal{L}_C^{(k)}$ calculates the difference between the resampling ground truth $x_{(k)}$ and the warped image $I_{i_{(k)}}^{warp}$ deformed by the optical flow $\vec{f}_{(k)}(inter_i, i)$ on the image $I_{i_{(k)}}^{inter}$ as follows:

$$\begin{aligned} \mathcal{L}_C^{(k)} = &\lambda_c \mathcal{L}_r(I_{i_{(k)}}^{warp} \odot (1-m_k), x_{(k)} \odot (1-m_k)) + \\ &\mathcal{L}_r(I_{i_{(k)}}^{warp} \odot m_k, x_{(k)} \odot m_k) + \\ &\mathcal{L}_{ssim}(I_{i_{(k)}}^{warp}, x_{(k)}) + \mathcal{L}_{per}(I_{i_{(k)}}^{warp}, x_{(k)}), \end{aligned} \tag{14}$$

where $\mathcal{L}_r$ represents the reconstruction loss, $\mathcal{L}_{ssim}$ represents the structural similarity loss, $\mathcal{L}_{per}$ represents the perceptual loss. All $w_k$ and $\lambda_c$ are constant coefficients. The detailed definition of each loss term is listed as follows:

**Reconstruct loss** $\mathcal{L}_r$ is used to measure the L1 loss between the images $x$ and $z$ at the pixel level:

$$\mathcal{L}_r = \|x - z\|_1. \tag{15}$$

**Structural Similarity Loss** $\mathcal{L}_{ssim}$ is used to calculate the difference in luminance, contrast, and structure of two images $x$ and $z$:

$$\mathcal{L}_{ssim} = 1 - SSIM(x, z) = 1 - \frac{(2\mu_x\mu_z + C_1)(2\sigma_{xz} + C_2)}{(\mu_x^2 + \mu_z^2 + C_1)(\sigma_x^2 + \sigma_z^2 + C_2)}, \tag{16}$$

where $\mu_x$ and $\mu_z$ are the average value of image $x$ and $z$, $\sigma_x$ and $\sigma_z$ are the standard deviation, $\sigma_z$ is the covariance, $C_1$ and $C_2$ are constants.

**Perceptual Loss** $\mathcal{L}_{per}$ is used to compare convolutional features between $x$ and $z$ and focuses on the perceived quality of the images

| Mask | Method | CREMIA | | | | CREMIB | | | | CREMIC | | | |
|------|--------|--------|--------|--------|--------|--------|--------|--------|--------|--------|--------|--------|--------|
| | | PSNR↑ | SSIM↑ | FSIM↑ | FID↓ | PSNR↑ | SSIM↑ | FSIM↑ | FID↓ | PSNR↑ | SSIM↑ | FSIM↑ | FID↓ |
| 10% | PEN-Net | 29.094 | 0.913 | 0.957 | 9.881 | 29.612 | 0.906 | 0.950 | 18.195 | 28.315 | 0.903 | 0.957 | 7.918 |
| | AOT-GAN | 30.454 | 0.920 | 0.968 | 6.909 | 30.595 | 0.908 | 0.967 | 15.550 | 29.311 | 0.905 | 0.963 | 11.749 |
| | CCPGAN | 30.982 | 0.940 | 0.976 | 4.511 | 29.397 | 0.903 | 0.972 | 9.295 | 28.319 | 0.901 | 0.967 | 7.793 |
| | SSF-Restoration | 34.430 | 0.971 | 0.980 | 5.738 | 32.993 | 0.950 | 0.969 | 13.356 | 32.611 | 0.959 | 0.973 | 7.526 |
| | Ours | **36.132** | **0.984** | **0.986** | **1.989** | **35.952** | **0.973** | **0.983** | **4.328** | **35.026** | **0.979** | **0.983** | **2.350** |
| 30% | PEN-Net | 23.529 | 0.680 | 0.906 | 16.468 | 24.469 | 0.678 | 0.900 | 33.903 | 22.888 | 0.660 | 0.899 | 22.388 |
| | AOT-GAN | 25.661 | 0.754 | 0.925 | 17.454 | 26.268 | 0.742 | 0.920 | 32.273 | 24.719 | 0.719 | 0.912 | 29.747 |
| | CCPGAN | 25.487 | 0.799 | 0.928 | 11.273 | 24.689 | 0.708 | 0.922 | 20.806 | 23.069 | 0.694 | 0.908 | 19.340 |
| | SSF-Restoration | 29.435 | 0.902 | 0.938 | 25.056 | 28.127 | 0.838 | 0.915 | 45.266 | 27.636 | 0.870 | 0.923 | 28.179 |
| | Ours | **31.091** | **0.951** | **0.957** | **5.389** | **30.753** | **0.910** | **0.949** | **11.656** | **30.754** | **0.944** | **0.949** | **6.157** |
| 50% | PEN-Net | 21.371 | 0.445 | 0.860 | 35.200 | 22.234 | 0.451 | 0.856 | 53.639 | 20.705 | 0.412 | 0.843 | 46.644 |
| | AOT-GAN | 22.512 | 0.546 | 0.823 | 25.513 | 23.136 | 0.516 | 0.822 | 41.317 | 21.537 | 0.484 | 0.798 | 36.471 |
| | CCPGAN | 23.164 | 0.662 | 0.885 | 16.323 | 22.338 | 0.513 | 0.876 | 29.839 | 20.832 | 0.502 | 0.856 | 30.363 |
| | SSF-Restoration | 27.031 | 0.832 | 0.898 | 47.570 | 25.946 | 0.736 | 0.866 | 74.264 | 25.353 | 0.782 | 0.874 | 49.629 |
| | Ours | **29.783** | **0.917** | **0.928** | **7.655** | **28.405** | **0.854** | **0.917** | **14.714** | **28.745** | **0.910** | **0.917** | **8.564** |

Table 1: Quantitative comparisons on existing image inpainting methods. ↑ means higher is better. ↓ means lower is better.

which is more fit for the human perception of image quality [13] :

$$\mathcal{L}_{per} = \sum_i \|p_i(x) - p_i(z)\|_1, \quad (17)$$

where $p_i(\cdot)$ represents the $i$th feature layer of the pre-trained VGG16 model.

In the second stage, we train the guided inpainting network with the loss function $\mathcal{L}_G$ of the generator and the loss function $\mathcal{L}_D$ of the discriminator as follows:

$$\mathcal{L}_G = \mathcal{L}_r(\hat{I}_i \odot (1 - \hat{m}), x \odot (1 - \hat{m})) +$$
$$\lambda_r \mathcal{L}_r(\hat{I}_i \odot \hat{m}, x \odot \hat{m}) + \lambda_{adv}\mathcal{L}_{G\_adv}(\hat{I}_i, x), \quad (18)$$

$$\mathcal{L}_D = \lambda_{adv}\mathcal{L}_{D\_adv}(\hat{I}_i, x), \quad (19)$$

where $\mathcal{L}_{G\_adv}$ and $\mathcal{L}_{D\_adv}$ are adversarial loss, $\lambda_r$ and $\lambda_{adv}$ are weighting coefficients.

**Adversarial Loss** $\mathcal{L}_{G\_adv}$ and $\mathcal{L}_{D\_adv}$ are used for optimizing generator and discriminator. The input image consists of two regions: the initial known region and the generated inpainting region. To distinguish the generated patches in the missing region from the real patches in the context, we use the discriminator that is constructed based on PatchGAN [12]. The adversarial loss for the generator is:

$$\mathcal{L}_{G\_adv} = -\mathbb{E}_{z \sim p_z}[D(z)]. \quad (20)$$

The hinge version of adversarial loss for the discriminator is:

$$\mathcal{L}_{D\_adv} = \mathbb{E}_{z \sim p_z}[max(0, 1 + D(z))] + \mathbb{E}_{x \sim p_{data}}[max(0, 1 - D(x))]. \quad (21)$$

## 4 EXPERIMENTS

FlowInpaint is compared with two single-image inpainting methods: GAN-based Unet-like network PEN-Net [28], multiple contextual feature fusion framework AOT-GAN [29], and two consecutive-image inpainting methods: feature "copying and pasting" architecture CCPGAN [31], frame interpolation based pipeline SSF-Restoration [11].

### 4.1 Dataset

To comprehensively evaluate the network performance, we generate synthetic data based on the commonly used datasets, CREMI[1]. It contains three image stacks, CREMIA, CREMIB, and CREMIC. Each stack has 125 consecutive images of shape $1250 \times 1250$ that are taken from the brain of an adult Drosophila melanogaster. This dataset is imaged by ssTEM and has the voxel spacing of $4 \times 4 \times 40$ nm per voxel.

For the anisotropic CREMI dataset, we select raw image sets $\{I_{i-1}^{raw}, I_i^{raw}, I_{i+1}^{raw}\}$ that contain three adjacent images spaced 40 nm apart, while for the isotropic EPFL dataset, we decide the space between $I_i^{raw}$ and its neighboring images $\{I_{i-1}^{raw}, I_{i+1}^{raw}\}$ based on specific experimental purpose. Then we randomly cut 2000 sets of images $\{I_{i-1}, I_i^{gt}, I_{i+1}\}$ with the shape $256 \times 256$ from the corresponding positions for training, while 100 sets for testing, and generate the simulated damaged images $I_i$ by applying the mask $\hat{m}$ on $I_i^{gt}$ as follows:

$$I_i = I_i^{gt} \odot (1 - \hat{m}) + \hat{m}. \quad (22)$$

Notably, the mask is of irregular shape to simulate the realistic damaged regions.

### 4.2 Implementation Details

FlowInpaint is implemented by PyTorch [18] and all the experiments are trained on the NVIDIA A100 GPU. The input images are of $256 \times 256$ size. For the refined reference module, we extract two feature pyramids with $K = 6$ and train the model with a learning rate schedule starting from 0.0001 and reducing the learning rate by half at 0.4M, 0.6M, 0.8M, and 1M iterations. For the guided inpainting module, we extract two multi-scale feature sets with $L = 6$ and adopt the learning rate of 0.0001. Both modules are trained by the Adam optimizer, where $\beta_1 = 0.5$, $\beta_2 = 0.999$. We set the weights $w_1 = 1$, $w_2 = 0.64$, $w_3 = 0.32$, $w_4 = 0.16$, $w_5 = 0.08$, $w_6 = 0.04$, $\lambda_c = 5$, $\lambda_r = 6$, and $\lambda_{adv} = 0.1$ in the loss functions.

---

[1]Dataset available at https://cremi.org/data/

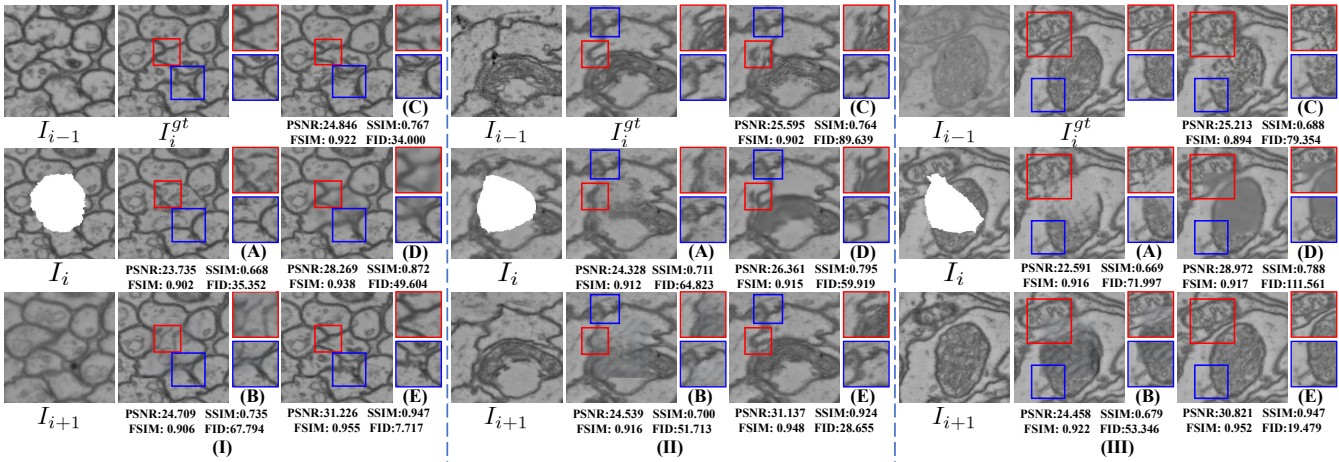

**Figure 4: The inpainted results with irregular damaged areas under 30% mask on the CREMI dataset. (A) PEN-Net, (B) AOT-GAN, (C) CCPGAN, (D) SSF-Restoration, and (E) Ours.**

### 4.3 Evaluation Metrics

The results of FlowInpaint are quantitatively compared based on the image quality assessment metrics: PSNR (Peak Signal-to-Noise Ratio), SSIM (Structural Similarity Index), FSIM (Feature Similarity Index Measure) [30], and FID (Fréchet Inception Distance) [10]. PSNR and SSIM evaluate the similarity between images based on pixel-value information. FSIM, on the other hand, analyzes image quality by comparing the structural similarity of images based on their local feature maps and calculating a weighted average of the results. FID measures the similarity between the feature distributions of the images based on the high-level perceptual features extracted through pre-trained InceptionV3 [21].

### 4.4 Quantitative Evaluation

Table 1 summarizes the quantitative evaluation results on three simulated testing data from the CREMI dataset. For CCPGAN, we use the images $I_{i-1}$ as its reference image to generate inpainted results. The single-image inpainting methods, PEN-Net and AOT-GAN, exhibit poor quantitative performance across various metrics when the masked area reaches or exceeds 30%. This is mainly because they only utilize similar structures in the undamaged regions to fill in the missing information, and a large masked region means an insufficient number of candidate patches. Compared to single-image approaches, the consecutive-image methods, CCPGAN and SSF-Restoration, perform better on the content-related metrics consistently under different masking ratios. They search for similar structural features from the neighboring images, not only from the known areas of the damaged section. Naturally, the biological structures along the axial direction provide more comprehensive and detailed contents, with most of the deformation being estimated numerically. Meanwhile, the more images for reference, the higher the precision for structure similarity measurement. That's why SSF-Restoration and FlowInpaint, using two neighboring sections, both have a large improvement compared to CCPGAN, which only inputs one reference image.

In image inpainting, the metrics solely measuring the difference between the corresponding pixels may result in fake results, which are easily judged visually. We adopt FSIM and FID two perceptual metrics in the feature domain to quantitatively evaluate the consistency between recovered contents and known areas. Benefits from the additional perceptual losses, FlowInpaint outperforms other methods across all datasets and masking situations. Notably, the SSF-Restoration always performs worse FID scores under the $\geq 30\%$ masked situation, which indicates the less structural consistency of inpainted content. This is largely due to its less-quality reference image, discussed in Section 4.6.2.

### 4.5 Visual Comparison

Figure 4 illustrates the visual comparison results. Since there is usually some deformation at the edges of irregular damages in the real world, we use a rectangular mask that can fully cover the damaged area based on its shape and then determine the specific size of the recovered area. The area that needs to be recovered for each sample in Figure 4 is approximately 30%. The generated content of the single-image inpainting methods, PEN-Net and AOT-GAN, is structurally unrelated to its neighboring images. Thus the single-image inpainting methods are completely inappropriate for solving the issue of missing section information in vEM. In Figure 4 (I), the brightness and clarity of $I_{i+1}$ are slightly lower than those of $I_{i-1}$. The CCPGAN's result utilizes highly correlated image features from $I_{i-1}$, thus its structure looks more similar to it and cannot depict the tissue morphology along the axial direction. The SSF-Restoration's result is not clear enough which may be influenced by the image properties of $I_{i+1}$. By contrast, our method generates clear and reasonable content. In Figure 4 (II), there are some noise points in $I_{i-1}$ and the significant structure change in consecutive images. The image features learned by CCPGAN from $I_{i-1}$ are not consistent with the known sectional information. Thus the factors of section thickness and rapid variation in serial sections significantly impact the effectiveness of CCPGAN. For SSF-Restoration, its result exhibits a distinct dislocation phenomenon at the edge of the damaged area. While our method is still the best. In Figure 4 (III), we can see the contrast and clarity of $I_{i-1}$ is extremely poor which greatly influences the results of CCPGAN and SSF-Restoration. CCPGAN

generates completely nonsensical content while SSF-Restoration outputs severely blurry content. However, even in structure-dense areas, our method can still produce relatively coherent results.

## 4.6 Ablation Studies

We conduct ablation experiments on the CREMI dataset to verify the effectiveness of each module in FlowInpaint.

*4.6.1 Reference Image Generation.* FlowInpaint adopts a two-stage reference image generation strategy to introduce the 3D axial structural continuity into the reference image. First, we remove the reference image generation and directly input two neighboring images to the guided inpainting module (denoted as GuidedGAN in Table 2). Without reference images, GuidedGAN though has high enough numerical metrics, it prefers to generate false structures, which are obviously shown in the visual example (Figure 5). Additionally, we replace the reference generation module with a frame interpolation approach used in SFF-Restoration and name it SFF+GuidedGAN. Owing to large deformation between the $I_{i+1}$ and $I_{i-1}$, SFF+GuidedGAN faces a great challenge to estimate a rational intermediate state once. Its inpainted image has obvious structural misplacement, indicated by a red arrow, and may result in a wrong structure label in the following segmentation task.

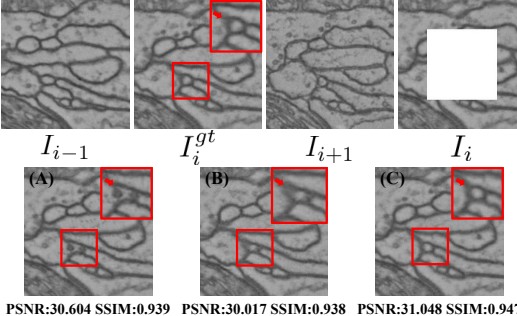

$$I_{i-1} \qquad I_i^{gt} \qquad I_{i+1} \qquad I_i$$

(A) PSNR:30.604 SSIM:0.939 FSIM:0.948 FID:9.324

(B) PSNR:30.017 SSIM:0.938 FSIM:0.949 FID:9.746

(C) PSNR:31.048 SSIM:0.947 FSIM:0.950 FID:8.676

**Figure 5: The visualization of different reference images. (A) GuidedGAN, (B) SFF+GuidedGAN, and (C) Ours.**

| Mask | Method | Metric | | | |
|------|--------|--------|------|------|------|
| | | PSNR↑ | SSIM↑ | FSIM↑ | FID↓ |
| 10% | GuidedGAN | 35.936 | 0.978 | 0.983 | 3.091 |
| | SFF+GuidedGAN | 34.447 | 0.971 | 0.982 | 3.185 |
| | Ours | **35.936** | **0.980** | **0.984** | **2.889** |
| 30% | GuidedGAN | 30.281 | 0.927 | 0.950 | 8.343 |
| | SFF+GuidedGAN | 29.190 | 0.901 | 0.945 | 8.524 |
| | Ours | **30.866** | **0.935** | **0.952** | **7.734** |
| 50% | GuidedGAN | 28.244 | 0.877 | 0.920 | 12.043 |
| | SFF+GuidedGAN | 26.829 | 0.823 | 0.913 | 11.867 |
| | Ours | **28.978** | **0.894** | **0.921** | **10.311** |

**Table 2: Ablation results on the reference image generation. ↑ means higher is better. ↓ means lower is better.**

*4.6.2 Guided Inpainting Module.* FAL can fully extract extensive contextual features from the reference information and the GAN-based framework can significantly boost the perceptual quality of the inpainted content. Table 3 summarizes the quantitative results of FlowInpaint with a version of FlowInpaint without FAL (Ours-FAL) and a version of FlowInpaint without GAN architecture (Ours-GAN). The introduction of FAL helps to explore and aggregate multi-scale features of multiple reference images, effectively lifting image quality at the pixel level (~2.0 improvement in PSNR and ~0.08 improvement in SSIM). Meanwhile, the adoption of GAN is a crucial key to perceptual quality. Without the help of discriminators, the network hardly yields acceptable restored content, especially the masking regions exceeding 30%.

| Mask | Method | Metric | | | |
|------|--------|--------|------|------|------|
| | | PSNR↑ | SSIM↑ | FSIM↑ | FID↓ |
| 10% | Ours-FAL | 34.029 | 0.968 | 0.981 | 3.227 |
| | Ours-GAN | 35.507 | 0.977 | 0.982 | 6.294 |
| | Ours | **35.936** | **0.980** | **0.984** | **2.889** |
| 30% | Ours-FAL | 28.675 | 0.889 | 0.943 | 9.207 |
| | Ours-GAN | 30.419 | 0.925 | 0.946 | 23.963 |
| | Ours | **30.866** | **0.935** | **0.952** | **7.734** |
| 50% | Ours-FAL | 26.291 | 0.813 | 0.908 | 12.367 |
| | Ours-GAN | 28.165 | 0.876 | 0.911 | 42.300 |
| | Ours | **28.978** | **0.894** | **0.921** | **10.311** |

**Table 3: Ablation results for guided inpainting module. ↑ means higher is better. ↓ means lower is better.**

## 5 CONCLUSION

In this paper, we propose FlowInpaint, a vEM image inpainting network to recover the missing information at the damaged regions under the guidance of two neighboring sections. Previous single-image inpainting methods only utilize 2D known cross-sectional information, and consecutive-image inpainting approaches adopt "copy-and-paste" strategy to fill similar structures from the neighboring images into the damaged region. These methods neglect the 3D structure changes along the axial direction and are quite sensitive to section thickness. On the contrary, FlowInpaint estimates optical flow from neighboring sections to construct 3D structural continuity. Meanwhile, the two-stage reference generation strategy effectively ensures strong stability under different thicknesses of samples. Compared to existing EM image inpainting networks, our method simultaneously considers the complexity of guaranteeing 3D structural continuity along the axial direction and preserving 2D structural consistency around the missing regions. Comprehensive experiments well demonstrate the superiority of FlowInpaint over existing methods in serial section inpainting. Further stability analysis reveals the applicability of FlowInpaint to different thicknesses of tissue sections and imaging situations, which are variable in real-world data.

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
