# OpenReview forum: "Serial section microscopy image inpainting guided by axial optical flow"
_acmmm.org/ACMMM/2024/Conference — MM2024 Poster_

### Official Review · Reviewer_Ljrz · 2024-05-21

**Rating:** 4
**Confidence:** 4

**Summary:**

This paper presents an optical flow-based serial image inpainting approach, which is able to recover the missing biological structures caused by section damage during the vEM imaging of three-dimensional (3D) cellular tissues. Specifically, a two-stage reference generation strategy is proposed to predict the intermediate state image from coarse to fine. Then, a GAN-based inpainting network is used to reconstruct the final structures. Experiments are conducted on  three CREMI datasets and demonstrate the superiority of the proposed method over existing inpainting models by evaluating PSNR, SSIM, FSIM, and FID.

**Strengths:**

1. The authors introduce a two-stage approach to recover damaged regions in serial sections, considering both 3D structural continuity and 2D image structural consistency. Incorporating optical flow estimation to generate intermediate images and to ultimately recover 3D image structures is an interesting design.
2. The proposed FlowInpaint exhibits superior performance in serial section inpainting compared to existing methods. Extensive ablation studies and analysis are also conducted that validates the effectiveness of different modules.

**Limitations:**

1. Please clarify the fairness of comparisons.
In the evaluation of experimental outcomes, traditional image inpainting methods have predominantly served as the baselines for comparison. Given the disparate task settings and potential inconsistencies between training inputs and outputs.
It is imperative that the authors provide a more detailed exposition of these comparison methods, particularly in their adaptation to biological image datasets.
2. Please explain the adaptation of optical flow estimation to the domain of biological images
The manuscript lacks a thorough explanation of the adaptation of optical flow estimation to the domain of biological images, despite its common usage in video media. As a cross-domain application, it's important for the author to meticulously consider the correlation and distinctions between successive frames in videos and the upper and lower images of 3D cellular structures. A more detailed analysis is necessary to elucidate these aspects and validate the applicability of the proposed approach.
3. Please clarify the difference to related articles.
This paper needs to explain the difference and improvement between an optical flow-based method raised by Gonzalez-Ruiz and PWC-Net.

**Suitability:**

2

---

### Official Review · Reviewer_RK75 · 2024-05-23

**Rating:** 4
**Confidence:** 3

**Summary:**

The authors propose a deep-learning based vEM image inpainting framework named FlowInpaint to recover the missing information at the damaged regions under the guidance of two neighboring sections, which considers both 3D structural continuity and 2D image structural consistency. The framework applies a two-stage reference image generation strategy to estimate plausible biological structures in the inpainted regions from coarse to fine. Experiments results on the public CREMI datasets show the superiority of the proposed method among other existing methods.

**Strengths:**

The framework applies a two-stage reference image generation strategy to estimate plausible biological structures in the inpainted regions from coarse to fine. The initial reference is interpolated between neighboring sections to preserve the axial 3D continuity. The subsequent refined reference module warps it to fit the realistic structures in the undamaged areas. A GAN-based image inpainting sub-network is implemented to generate final seamless results.
Experiments results on the public CREMI datasets show the superiority of the proposed method among other existing methods.

**Limitations:**

1. Missing some necessary experiments.
e.g.,
a) Have the authors ever tested the robustness of the proposed method when the distance between adjacent images is larger (e.g., 80nm?).
b) Have the authors tested the efficiency of the proposed method on much larger sections?
c) What if two adjacent images have artifacts at the same area?
d) The performance of the proposed method using different number of levels k is not tested.

2. Some details about the proposed method are not clearly described
a) What is the isotropic EPFL dataset mentioned in Line 670?
b) How to choose the number of levels (k)?

**Suitability:**

3

---

### Official Review · Reviewer_tiGA · 2024-05-27

**Rating:** 2
**Confidence:** 3

**Summary:**

1. This paper proposes a novel EM image inpainting framework to recover damaged regions in serial sections.
2. This paper designs a two-stage reference image generation strategy,  to estimate biological structures in the inpainted regions in a multi-scale manner.
3. This paper adopts a GAN-based image inpainting sub-network to generate final results.
4. This paper conducts comprehensive experiments based on the CREMI datasets.

**Strengths:**

1. This paper proposes a new idea to recover damaged regions in ssEM data, while considering both 3D structural continuity and 2D image structural consistency.
2. This paper's design is easy to understand, and details have been discussed comprehensively.
3. This paper's writing and figures are relatively clear.

**Limitations:**

1. Relevance to multimedia is not clearly stated. I think the paper about electron microscopy image processing may be more suitable for MICCAI or ISBI conferences.
2. I think serial section image inpainting is something similar to video inpainting, but this paper misses the related discussion, can you describe the (dis)similarity between ssEM image inpainting and video inpainting? And can recent advanced video inpainting methods (such as "Inertia-guided flow completion and style fusion for video inpainting") be utilized to solve this problem?
3. This paper holds an assumption "we assume that the biological structures are transformed smoothly along the axial direction", but in real cases, there are many unregistered serial section EM datasets. Recent papers (such as "Fast and Accurate Electron Microscopy Image Registration with 3D Convolution") have discussed the learning-based registration methods for the ssEM dataset, and I think you should add the discussion with EM registration methods. In other words, you should consider the pre-registration operation before your methods, or you just combine the registration with your methods for the real cases. Otherwise, your method is very limited.
4. All your experiments are conducted on the CREMI datasets, but I think CREMI datasets are not enough. On the one hand, CREMI datasets do not involve unregistered cases, on the other hand, CREMI datasets do not have diverse corrupted distortions compared with the real cases. So I think you should add more experiments from other real ssEM datasets, which only contain the corruptions but do not have the ground-truth image, and you can give the visualization results for the comparison.
By the way, the experiments about "the isotropic EPFL dataset" mentioned in the paper do not seem to appear in the paper.
5. The PWC-Net optical flow structure adopted in the paper is not advanced now, can other advanced optical flow structures bring more improvements?
6. How about the generalization ability of your method, such as the trained model from CREMI dataset can directly apply to other (different) EM datasets?

**Suitability:**

2

---

### Official Review · Reviewer_zU8J · 2024-06-02

**Rating:** 3
**Confidence:** 3

**Summary:**

This paper proposes an axial optical flow-guided microscopy image inpainting method, utilizing a GAN-based two-stage strategy. The structure is comprehensive, making it a commendable paper. However, improvements can be made in the experimental section, specifically in including comparative algorithms from the past two years and analyzing the loss functions.

**Strengths:**

The paper is well-structured, logically coherent, and rich in experiments. The proposed method demonstrates excellent qualitative and quantitative performance in the microscopy image inpainting problem.

**Limitations:**

1. The text and images in Figure 2 overlap; the layout could be further optimized.
	2. There are several deficiencies in the experimental section:
		○ The necessity of using multiple loss functions, such as SSIM loss and perceptual loss, should be experimentally validated in the paper.
		○ The parameter count and computational load of different methods should be indicated. Additionally, more recent and excellent inpainting methods from the past two years should be included in the comparison.
		○ Since some methods are not specifically designed for microscopy image inpainting, direct comparisons may lead to unfairness.
		○ Using generative methods may harm pixel-level metrics such as PSNR and SSIM while benefiting perceptual metrics like FID, which is usually related to the weighting of loss functions. The authors should conduct relevant experiments to address this

**Suitability:**

2

---

### Meta-Review · Area_Chair_nQVD · 2024-06-27

**Recommendation:** Accept (Poster)
**Confidence:** 4

**Metareview:**

After rebuttal, this paper received mixed ratings: 3 BAs and 1 BR. It appears that the rebuttal has done a good job of resolving reviewers' concerns. I would recommend acceptance, and ask the authors to consider improving their paper before the camera-ready due by incorporating the following comments: "I think more qualitative results about the impact of registration should be provided, and the generalization results are also important."